# Integrative Strategy of Testing Systems for Identification of Endocrine Disruptors Inducing Metabolic Disorders—An Introduction to the OBERON Project

**DOI:** 10.3390/ijms21082988

**Published:** 2020-04-23

**Authors:** Karine Audouze, Denis Sarigiannis, Paloma Alonso-Magdalena, Celine Brochot, Maribel Casas, Martine Vrijheid, Patrick J. Babin, Spyros Karakitsios, Xavier Coumoul, Robert Barouki

**Affiliations:** 1Inserm UMR S-1124, Université de Paris, 75006 Paris, France; xavier.coumoul@parisdescartes.fr (X.C.); robert.barouki@parisdescartes.fr (R.B.); 2HERACLES Research Center on the Exposome and Health, Aristotle University of Thessaloniki, Center for Interdisciplinary Research and Innovation, 57001 Thessaloniki, Greece; denis@eng.auth.gr; 3Instituto de Investigación, Desarrollo e Innovación en Biotecnología Sanitaria de Elche (IDiBE), Universidad Miguel Hernández, 03202 Elche, Spain; palonso@umh.es; 4Centro de Investigación Biomédica en Red de Diabetes y Enfermedades Metabólicas Asociadas (CIBERDEM), 28029 Madrid, Spain; 5Institut National de l’Environnement Industriel et des Risques (INERIS), Unité Modèles pour l’Ecotoxicologie et la Toxicologie (METO), Parc ALATA BP2, 60550 Verneuil en Halatte, France; Celine.BROCHOT@ineris.fr; 6ISGlobal, 08003 Barcelona, Spain; maribel.casas@isglobal.org (M.C.); martine.vrijheid@isglobal.org (M.V.); 7Universitat Pompeu Fabra (UPF), 08003 Barcelona, Spain; 8CIBER Epidemiología y Salud Pública (CIBERESP), 28029 Madrid, Spain; 9Department of Life and Health Sciences, University of Bordeaux, INSERM U1211, MRGM, F-33615 Pessac, France; patrick.babin@u-bordeaux.fr; 10Enve.X, 55133 Thessaloniki, Greece; spyros@eng.auth.gr; 11Service de Biochimie métabolomique et protéomique, Hôpital Necker enfants malades, AP-HP, 75015 Paris, France

**Keywords:** endocrine disruptors, metabolism, integrative approach, predictive toxicology, metabolic disorders, obesity, non-alcoholic fatty liver diseases, non-animal testing, adverse outcome pathways, computational systems toxicology

## Abstract

Exposure to chemical substances that can produce endocrine disrupting effects represents one of the most critical public health threats nowadays. In line with the regulatory framework implemented within the European Union (EU) to reduce the levels of endocrine disruptors (EDs) for consumers, new and effective methods for ED testing are needed. The OBERON project will build an integrated testing strategy (ITS) to detect ED-related metabolic disorders by developing, improving and validating a battery of test systems. It will be based on the concept of an integrated approach for testing and assessment (IATA). OBERON will combine (1) experimental methods (in vitro, e.g., using 2D and 3D human-derived cells and tissues, and in vivo, i.e., using zebrafish at different stages), (2) high throughput omics technologies, (3) epidemiology and human biomonitoring studies and (4) advanced computational models (in silico and systems biology) on functional endpoints related to metabolism. Such interdisciplinary framework will help in deciphering EDs based on a mechanistic understanding of toxicity by providing and making available more effective alternative test methods relevant for human health that are in line with regulatory needs. Data generated in OBERON will also allow the development of novel adverse outcome pathways (AOPs). The assays will be pre-validated in order to select the test systems that will show acceptable performance in terms of relevance for the second step of the validation process, i.e., the inter-laboratory validation as ring tests. Therefore, the aim of the OBERON project is to support the organization for economic co-operation and development (OECD) conceptual framework for testing and assessment of single and/or mixture of EDs by developing specific assays not covered by the current tests, and to propose an IATA for ED-related metabolic disorder detection, which will be submitted to the Joint Research Center (JRC) and OECD community.

## 1. Introduction 

Evidence for environmental and human health effects of endocrine disruptors (EDs) has increased considerably over the last few years. Notably, several health outcomes have now been associated with the exposure to endocrine-disrupting chemicals, such as certain cancers (e.g., hormone-dependent tumors), neurocognitive and neurodevelopmental disorders, infertility, immune diseases, and allergies, to mention but a few outcomes. Metabolic diseases including diabetes, insulin resistance, obesity, steatosis, metabolic syndrome, chronic liver diseases, etc., are among the most prominent health outcomes of human exposure to EDs [1]. Experimental and computational studies confirmed the association between certain EDs and metabolic diseases and provided additional mechanistic insight. The economic impact of ED use in the European Union (EU) was estimated recently by a group of scientists to be around EUR 150 billion per year in the EU and this is probably an underestimated amount [2]. 

Metabolic diseases were estimated to be the second most costly health outcome (after intelligence quotient (IQ) loss) [3]. In the United States (US), the costs are likely more than the double this amount. It is therefore critical to develop tools to robustly identify EDs to protect citizens in the EU and worldwide and also to limit significant costs associated with those compounds. 

Regulatory measures have been taken in the EU and member states restricting the use of certain EDs. However, there are still critical issues to be addressed to ensure sufficient protection of EU citizens. (1) Regulatory decisions are based on criteria for the definition of EDs and this has been controversial in the EU. Recently, those criteria were agreed upon in the case of biocides and pesticides (https://ec.europa.eu/health/endocrine_disruptors/overview_en). Following these decisions, it is now critical to establish validated tests addressing those criteria and allowing regulatory authorities to carry out a risk assessment and to take relevant decisions. (2) In some cases, substituents to regulated compounds have been used, but there are still uncertainties concerning the safety of those substituents and more generally concerning new compounds or poorly studied compounds; this is the case for example of bisphenol S (BPS) and bisphenol F (BPF) as substituents to bisphenol A (BPA), DINCH as a substituent to phthalates and GenX as a substituent to PFOA. (3) Current OECD endocrine in vitro tests include estrogen or androgen receptor binding affinity, estrogen receptor transactivation, androgen transactivation assay and in vitro steroidogenesis (https://www.oecd.org/env/ehs/testing/oecdworkrelatedtoendocrinedisrupters.htm). There is indeed a lack of robust tests predicting metabolic outcomes of exposure to chemicals. The limitations are related to species-dependent metabolic pathways and the uncertain value of certain simple tests such as receptor activation. New developments in ED testing are warranted, as recently outlined by OECD and Joint Research Center (JRC) study groups. 

In order to shift from current animal-based testing towards the application of emerging technologies (incl. metabolomics as well as systems biology), to reduce time and costs, the European Union REACH program (Registration, Evaluation, and Authorization of Chemicals, REACH) supports the increased integrative use of in vitro and in silico testing methods, which are in line with the National Academy of Sciences publication Toxicity in the 21st Century: A Vision and a Strategy (National Research Center (NRC), 2007) [4]. Moreover, the development of an animal-free strategy is aligned with the EU endorsement of replacement, reduction, and refinement (3Rs) principle aiming to minimize animal use in toxicological research, including ED research, because of ethical concerns. 

The main objective of the present paper is to introduce the newly established EURION cluster, and to provide an overview of the strategy of the five-year OBERON project, that officially started on 1 January 2019, to develop new alternative test systems for the identification of EDs related to metabolic disorders.

## 2. The EU Initiative

With the new regulations concerning EDs in certain sectors and within REACH in Europe, it became critical to improve the capacity to test those compounds and to identify the major outcomes. Despite the significant progress made in the identification of EDs during the last two decades which are captured in OECD guideline tests, several gaps in the testing strategy remain. The European Commission, and particularly the directorate-general (DG) environment and the JRC, conducted a survey to identify those gaps and organized a workshop dedicated to this particular issue (report on DG Environment workshop: https://op.europa.eu/en/publication-detail/-/publication/6b464845-4833-11e8-be1d-01aa75ed71a1/language-en; report on the JRC survey: https://publications.jrc.ec.europa.eu/repository/handle/JRC106244). There was a consensus within the group on the major health outcomes that needed further testing or improvements in current tests. Those outcomes were thyroid-related neurodevelopmental effects, metabolic dysfunctions, reproductive effects and growth and developmental effects. Based on those conclusions, the EC launched a call for research on new ED testing in these particular fields and eight projects were selected covering those health outcomes. These projects are grouped in a cluster called EURION, which aims at developing synergies, supporting interactions and leading working groups on priority issues including pre-validation and applicability of the developed tests (https://eurion-cluster.eu/). OBERON is one of the successful projects and its focus is on metabolic outcomes (https://oberon-4eu.com/).

In addition to the foreseen regulatory outcomes, there are some good reasons for coordinating EDs testing research in the EU within a cluster such as EURION. Indeed, while the different projects have their objectives, there are clear overlaps that the cluster can capture and coordinate. For example, thyroid hormones have an impact both on metabolism and neurodevelopment. In addition, metabolism is implicated in all physiological systems and its disruption may be relevant to other targets addressed by the different projects. Therefore, EURION will have a coordinating role but also to a certain extent a role in integrating different outcomes from the constituting projects. 

## 3. The OBERON Project 

Despite the high incidence of metabolic diseases, most available mammalian in vivo assays do not address ED-related mechanisms of action, and the specific endpoints associated with metabolic disorders are missing. Furthermore, metabolic pathway dysregulations are not only involved in the development of metabolic disorders as such (diabetes, obesity, etc.), but they are also critical for the development of a number of other outcomes such as cancer, immune diseases, and neurological diseases. Thus, the availability of relevant tests for metabolic dysregulation may have impacts that are much wider than those related to metabolic diseases. 

*The OBERON Vision*. The OBERON project aims at providing a series of new effective and validated tests, which are essential for industries and regulatory needs, to support risk assessment of putative ED chemicals in relation to metabolic disorders. By its multidisciplinary strategy and the establishment of an integrated approach for testing and assessment (IATA), the ultimate aim of OBERON is to provide more accurate predictive testing by developing a battery of novel, easy to use, effective and validated test systems, combining different experimental and computational strategies. The battery will include the most relevant developed/improved test systems in a decisional tree and should ultimately be used for regulatory purposes regarding ED assessment in relation to metabolic disorders (Figure 1). 

In order to develop such a test system, the OBERON vision relies on a multidisciplinary approach supported at all stages by data analysis and computational studies. The different tests will be built on ‘realistic exposure to chemical substances’ using data from well-characterized on-going epidemiology and human biomonitoring studies (biomarkers concentrations in cord blood, peripheral blood or urine samples) as well as the literature. OBERON tests will be developed based on a panel of ten EDs (Table 1), selected because of their widespread use, potential toxicity and production in large quantities worldwide (millions of tons annually). Most of them are of regulatory interest in Europe (e.g., BPA, di(2)ethylhexyl)phthalate (DEHP)) and others are well-known EDs that can be used as “known-controls” (heavy metals, dichlorodiphenyldichloroethylene (DDE)). Substituents to these compounds have also been selected (e.g., BPS and BPF) because, although they have been considered safer than the original compounds, they have similar properties and their health effects are still largely unknown. Data generated within OBERON, combining cell biology, zebrafish, omics technologies, and systems biology, will allow improving knowledge on EDs. Novel mechanistic information will be established, allowing the development of uncharacterized adverse outcome pathways (AOPs), as well as AOP networks, and identification of novel biomarkers to improve risk assessment frameworks for human health effects. To reflect a ‘real life’ exposure, birth cohorts and other epidemiological studies’ information will be used to determine a set of case studies that will be applied to a set of new easy-to-implement combined complementary methods (in vitro, in vivo, in silico). Therefore, to reach its aims and establish an innovative IATA, OBERON has identified several objectives.

### 3.1. Objective 1: Integration of Epidemiology and Human Biomonitoring Studies with ED Test Systems for Metabolic Disorders

OBERON aims at using relevant human data in the development of the test battery. Epidemiological evidence and human biomonitoring studies are currently scarcely used for regulatory purposes, partly due to the small sample size of some of the previous studies and to the difficulty in translating exposure levels in experimental studies to human exposure levels [5]. With respect to metabolic disorders, population studies usually rely on global outcomes such as body mass index (BMI) and blood pressure due to the difficulty in obtaining more specific outcomes such as comprehensive metabolic profiles. The interpretation of previous studies has also been limited due to large uncertainties in exposure assessment for EDs with short half-lives [6]. These studies usually rely on a few spot urine samples collected during the critical periods of development (e.g., pregnancy) which only give us information on short-term exposure; this leads to an underestimation of the health effects associated with ED exposure [7,8]. Studies with a larger sample size and with a more comprehensive assessment of exposure to non-persistent EDs and metabolic disorders are needed to integrate epidemiological evidence into ED test systems in a more meaningful way. 

Within OBERON, existing data from various European human cohorts will be analyzed in order to provide epidemiological evidence on associations between EDs and metabolic disorders (Table 2). We will work with large longitudinal cohort datasets with available individual exposure measurements for EDs and data on body composition, blood pressure, lipid profiles, liver enzymes, glucose, and insulin. Recent studies with multiple repeated biomarker assessments will be integrated in order to reduce exposure measurement errors for non-persistent EDs. Various time windows will also be considered, including developmental periods of high vulnerability, windows offering prevention opportunities (pregnancy-childhood-adolescence), and adulthood. By using data from five pregnancy cohorts, we will be able to investigate the effects of in utero exposure to EDs during childhood and adolescence. Some of these cohorts started more than a decade ago and others more recently, thus reflecting past and current exposure to EDs. We will also use data from adult cohorts in the Czech Republic to investigate the association of exposure to non-persistent EDs and anthropometric and metabolic outcomes in adults. Lastly, metabolomic profiles will be determined in adolescents and adults to understand the biological pathways involved in EDs exposure.

Currently, tests for predicting human toxicity of EDs are based on animal studies, which do not always reflect human exposure and response. In order to develop ED tests with realistic doses and relevant exposure, data collected from some of Europe’s best characterized prospective birth cohorts and human biomonitoring studies will be used to characterize levels of exposure to selected EDs. These concentrations will serve to determine the ED doses used in the in vivo (Objective 2) and in vitro (Objective 3) studies and to estimate the internal tissue dose of real-life EDs exposures by using physiologically based pharmacokinetic models (Objective 4).

### 3.2. Objective 2: Development of Whole Organism Test Systems to Identify EDs Implicated in Metabolic Disorders

The relevance of the Zebrafish model. The in vivo test systems developed in OBERON use zebrafish (*Danio rerio*), a well-recognized and flexible biological model system that fits between in vitro models and mammalian rodent models. In toxicology research, it enables studying and modeling the toxicity from molecular initiating events (MIEs) to alterations in organismal health. Zebrafish can be used for high-throughput EDs toxicity testing, allowing for quick, large scale screening similar to in vitro assays [14]. The strength of the model lies with highly conserved organ systems and metabolic pathways leading to a good concordance between zebrafish and mammalian models in evaluating chemicals for organ system toxicology [15]. Some features of zebrafish development are exploited in toxicity testing. Indeed, they are optically translucent in the early stages of ex vivo development, allowing for easy exposure to EDs and investigation of their mechanisms of action. The zebrafish model presents many methodological advantages and extensive collections of useful resources are available (https://zfin.org/). These include tools applicable in vivo, e.g., whole-mount imaging, methodologies for modulating gene expression, behavioral tests to examine changes in motor activity, and simplified simultaneous chemical/drug testing on a large number of animals. Library of small molecules including potential EDs can be arrayed in multi-well plates containing zebrafish embryos or larvae and metabolic disorders may be initially identified using phenotype-based approaches in the context of a whole organism. 

Based on all those properties, the zebrafish model appears to be an easily accessible, inexpensive, experimental model for in vivo screening of potential EDs that must be in concordance with epidemiological studies and in vitro tests (Objectives 1 and 3) to be integrated with modeling approaches (Objective 4) and computational systems biology (Objective 5). Receptor tests for obesogens. Obesogens are a specific class of EDs that promote obesity by altering adipocyte tissue development, lipid homeostasis, and hormonal physiology. It was recently demonstrated that obesogens may exert some of their biological effects via a nuclear receptor (NR)-dependent pathway, providing a direct molecular link between environmental ED exposure, endocrine-disrupting effects, and adipose tissue development [16]. Obesogens increase obesity through a variety of potential mechanisms, including the activation of peroxisome proliferator-activated receptor gamma (PPARγ), a key regulator of adipogenesis [17,18]. Assessing the activation of NR by EDs is certainly relevant in a screening strategy. Pathways important for adipogenesis and lipid metabolism are conserved between mammals and teleost fish [19]. In OBERON, a cell line stably transfected with zebrafish PPAR subtypes and luciferase as a reporter gene will be used in order to quantify the biological activity of substances towards zebrafish PPAR subtypes. These data will be compared with already published data, e.g., the literature, databases such as ToxCast, to identify potential similarities or differences between zebrafish and humans.The zebrafish obesogenic test. Activation of PPARγ is not sufficient to classify a compound as an obesogen. We therefore need a complementary whole-organism mechanism-based assay for screening substances acting as potential obesogens. The semitransparent zebrafish larvae with well-developed white adipose tissue offer a unique opportunity for studying the effects of chemicals on adipocyte biology and whole-organism adiposity in a vertebrate model [20]. The use of integrative methods is of prime importance for screening substances interfering with adipogenesis. We developed a simple short-term in vivo assay called the zebrafish obesogenic test (ZOT) and used it to examine the effects of diet, drugs, and environmental contaminants, singly or in combination, on white adipose tissue (WAT) dynamics in zebrafish larvae [20]. The ZOT was proven to be very useful in characterizing obesogenic or anti-obesogenic substances including EDs [21] and their mechanism of action [22].The zebrafish liver disease test. Non-alcoholic fatty liver disease (NAFLD) is a subset of liver disease beginning with simple steatosis which can evolve into steatohepatitis. NAFLD constitutes a public health priority as it can ultimately progress towards more severe and irreversible stage such as cirrhosis or hepatocellular carcinoma. NAFLD is prevalent in countries that consume a western diet. However, its actual pathogenesis remains elusive and, in this context, exposure to xenobiotics, especially EDs, has been suspected [23]. Zebrafish larva liver displays similarities to human models in the assessment of chemicals effects on steatosis and its transition to steatohepatitis [24]. OBERON is involved in the optimization of NAFLD test methods using zebrafish larva in order to screen potential EDs for their involvement in liver steatosis and its transition to steatohepatitis.

All these studies, together with analytical chemistry and omics analyses, will produce data that will be exploitable (1) to perform modelling approaches (e.g., pharmacokinetic (PK) and physiologically based pharmacokinetic (PBPK) models (Objective 4), (2) to develop integrative systems biology models (Objective 5), and (3) to propose novel AOPs (Objective 5). For example, cross-omics studies that will involve lipidomics, metabolomics and transcriptomics signatures of exposure and pathologies will be carried out, allowing the identification of potential MIEs and key events (KEs). 

Following those studies, intra- and inter-laboratory reproducibility of ZOT and NAFLD test methods will be assessed within the OBERON consortium.

### 3.3. Objective 3: Development of Human-Relevant In Vitro Test Systems to Identify EDs Involved in Metabolic Disorders

In vitro models have been successfully used to study physiological and pathophysiological processes relevant to human diseases. They are currently considered central to chemical tests and screening, as well as chemical safety assessments and they are an essential component of the application of the 3R principle [25]. They are also less expensive and faster than animal-based experiments. While the lack of complexity compared to an entire organism has been a major criticism, a significant advance in cell culture techniques has allowed the development of models that better match in vivo conditions [26].

Over the last two decades, the OECD has revised existing tests and developed new test guidelines for the screening and testing of ED chemicals. The test collection, as well as some guidelines for data interpretation are contained within the OECD Conceptual Framework for the Screening and Testing of Endocrine Disrupting Chemicals (http://www.oecd.org/publications/guidance-document-on-standardised-test-guidelines-for-evaluating-chemicals-for-endocrine-disruption-2nd-edition-9789264304741-en.htm). Although the ultimate goal of the framework is to provide harmonized and validated methods, it should be noted that it is not designed to be a testing strategy and does not take into consideration the evaluation of exposure. The guidelines tests focus on estrogenic, androgenic, thyroid hormone and steroidogenesis pathways. However, many other pathways that are important for ED-induced disorders like diabetes, obesity, hepatic steatosis or metabolic syndrome are not assessed by the current validated methods. 

OBERON will improve and develop human in vitro cellular models representing the main organs and tissues involved in the etiology of metabolic diseases (liver, endocrine pancreas, and white adipose tissue). These systems will include advanced 2D-models but also complementary 3D tissue models and organoids. The chemical concentrations to be used in cellular models will be calculated following the toxicokinetic prediction of tissue-level concentrations in human studies. The liver system. The human hepatocellular carcinoma cell line HepaRG (HepaRG) cell line is constituted of human hepatic progenitor cells which, under appropriate culture conditions, are able to give rise to two different types of cells, one expressing differentiated hepatocytes functions and the other biliary cells [27]. These two types of cells maintain significant liver functions, such as detection of xenobiotics and subsequent induction of xenobiotics-metabolizing enzymes including cytochrome P450 (CYP450) and transport activities. HepaRG cells can be grown in 2D, either in the absence or in the presence of inflammatory cells, which represents a model system to study inflammatory metabolic diseases. They can also be grown in 3D conditions, leading to organoids which better mimic in vivo conditions. The OBERON project will also use the human hepatocellular carcinoma cell line HepG2 (HepG2) cell line or the immortalized human hepatocytes (MIHA) cells, a model of immortalized human hepatocytes which have been successfully applied both in cytotoxicity and certain mechanistic studies, covering a wide range of function of liver cells. Primary cells (hepatocytes and pre-adipocytes) will also be used in order to strengthen data generated from closely related differentiated cell lines.The pancreas system. We will use the murine pancreatic β and α-cell lines (mouse insulinoma 6 (MIN6), alphaTC1 Clone 9 (αTC1.9)) and a human pancreatic β-cell line (EndoC-βH1). Experiments will be performed first in the murine secreting cell models, which have been previously demonstrated to be valuable tools for toxicological studies and screening purposes, and then in the human model which displays an enhanced β-cell phenotype and genomic stability over 100 passages [28].The adipose tissue. We will use a) human multipotent adipose-derived stem cells (hMADS), a mesenchymal stem cell line from human adipose tissue of young male and female donors which is also able to convert into functional brown-like adipocytes or b) human pre-adipocyte Simpson–Golabi–Behmel syndrome (SGBS) cell line which originates from adipose tissue and is able to proliferate for up to 50 generations with retained capacity for adipogenic differentiation. So far, these cells have been used for a number of studies on adipose differentiation, adipocyte glucose uptake, lipolysis, apoptosis, regulation of expression of adipokines, and protein translocation [29]. The cells could be efficiently differentiated in the presence of PPARγ agonists (including some EDs) and the absence of serum. Importantly, many of the cell lines used in our studies can differentiate in vitro. This will allow us the opportunity not only to address metabolic dysfunctions in differentiated cells but also to assess the capacity of EDs to alter cell differentiation, a setting that could mimic the developmental effects observed in epidemiological studies and that could be mediated by epigenetic regulation.

Therefore, within OBERON, several cellular systems will be tested in order to select those that most faithfully represent human and in vivo responses to ultimately include them in ITS. The rationale for the selection will be based on the study of relevant metabolic endpoints which precede and/or exacerbate metabolic disorders (e.g., altered fatty acid synthesis and β-oxidation, enhanced adipogenesis, impaired insulin secretion) as well as large-scale metabolomics and transcriptomics analysis. The use of a specific omics methodology will provide a greater understanding of the mechanisms and biomarkers underlying the potential associations between the exposure to EDs and the progress to metabolic diseases. Furthermore, the analysis of the metabolic signatures occurring after EDs exposure will help us to determine the most relevant outcomes to be selected in a final tiered test.

In vitro data will be further linked with in vivo data obtained in zebrafish models (Objective 2) as well as epidemiology and human biomonitoring studies (Objective 1) and advanced computational models (in silico and computational systems biology, Objectives 4 and 5) in order to confirm the relevance of the in vitro test systems developed. Test methods that show acceptable performance in terms of relevance will be selected for the inter-laboratory validation process within the OBERON consortium.

### 3.4. Objective 4: Providing Computational Models to Help Prioritization of EDs

While in silico testing cannot substitute for in vivo testing, it can help focus on particular substances and targets to allow priority-setting for more efficient testing [30,31,32]. Several international groups (e.g., European food safety authority (EFSA), European partnership for alternative approaches to animal testing (EPAA), European chemical agency (ECHA) etc.) promote the development of testing strategies for regulatory needs by using complementary methods based on in vitro and computational tools in combination with integrative systems biology and human exposure. Within OBERON, advanced computational models will be established to help prioritization of compounds based on their ED potential towards metabolic disorders. Quantitative structure-activity relationship (QSAR) models have a valuable predictive potential in the field of endocrine disruption because of the crucial role played by the molecular interactions between nuclear receptors and chemicals in defining and triggering the initiating events of endocrine disruption [33,34]. From a practical and regulatory point of view, the relevance of structure-activity approaches in identifying chemicals associated with a disrupting hazard has been especially and extensively established for the estrogen receptor for a wide range of chemical classes, proving the reliability of QSAR models as predictive tools for endocrine disruption in priority setting contexts [35]. In OBERON, existing QSAR models will be optimized based on new experimental data generated in the project (Objectives 2 and 3) and from literature to characterize ED induced effects on metabolism and obesity. Novel QSAR models will also be established based on (1) priorities identified by AOP network investigations (Objective 5), and (2) events already identified, such as the thyroid function and nuclear receptors. The integration of multiple models will also be explored in a weight of evidence strategy. All QSAR models will be implemented within the VEGA platform (https://www.vegahub.eu/).Physiologically based pharmacokinetic (PBPK) models are continuously gaining ground in regulatory toxicology, describing in quantitative terms, the absorption, metabolism, distribution, and elimination (ADME) processes in the human body, with a focus on the effective dose at the expected target site. The need for the widespread use of PBPK models development is amplified by the continuously increasing scientific and regulatory interest about aggregate and cumulative exposure; PBPK models translate external exposures from multiple routes into internal exposure metrics, addressing the effects of exposure route in the overall bioavailability. These models also address increased vulnerability to EDs during the critical developmental windows of susceptibility, such as embryonic and fetal stages, and childhood [36]. This vulnerability could be due to the epigenetic remodeling occurring during such periods, highlighting the string relevance of epigenetic regulations. It could also be due to the developmental physiology and exploratory behaviors of children, but also to the immaturity of their xenobiotic detoxifying processes or different adiposity during the early stages of development. Within OBERON, we will develop and apply human PBPK models for EDs that cover the whole life of an individual including prenatal life. QSAR models will be developed to support the parametrization of the PBPK models for some ADME processes, e.g., partitioning into the tissues, metabolic clearance, and placental transfers. PBPK models will be used to reconstruct complex exposures for the various EDs from biomonitoring data (Objective 1), allowing us to estimate the internal dose of real-life EDs exposures both on the organism level and in target tissues. This is of particular importance considering the complex toxicokinetic and toxicodynamic behavior of EDs that influences their bioavailability and toxic potency. This approach will also allow reconstructing the exposure in-between biomarker measurement time points, especially at critical periods. The knowledge of the internal exposures in the target tissues will be critical for extrapolating effects observed in the in vitro systems (Objective 3) to humans [37,38]. Dedicated toxicokinetic models for the in vitro systems used in OBERON will be built in order to provide cellular concentrations to be used for dose-response or systems biology modeling and integration within AOPs (Objective 5).

### 3.5. Objective 5: Establishment of an ITS and Capture of Mechanistic Effects of EDs on Metabolic Disorders

To identify the perturbations induced by the chemicals related to OBERON at different levels of biological organization, a cross-omics paradigm encompassing transcriptomics, epigenetics, and lipidomics/metabolomics signatures will be done. 

Transcriptomics analysis will be performed in both in vivo (zebrafish), and in vitro models, while epigenetic analysis will be carried out only using in vitro models. The aim is to identify the toxicity mechanisms associated with transcriptional and post-transcriptional changes in gene expression that can be causally linked to toxic outcomes. For the zebrafish model, transcriptomics analyses will start with whole exome sequencing methodology, and relevant targets will be confirmed by quantitative polymerase chain reaction (qPCR). All samples will be analyzed with both DNA microarray and microRNA array using the Agilent technology. Gene interaction analysis will be performed using R packages and information from various data sources. Metabolomics, including lipidomics analysis, will be carried out in human, in vivo (zebrafish) and in vitro samples. Metabolomics represents an effective approach for determining the activation of multiple AOPs and expanding knowledge on these AOPs relevant to complex exposures. This includes the identification of previously undefined key events by measuring changes in endogenous metabolites involved in a wide variety of biochemical pathways and associating these changes with exposure to combinations of xenobiotics. Existing protocols for human plasma, in vivo, and cellular model extracts will be used, with state-of-the-art equipment that includes gas chromatography-high resolution mass spectrometry (GC-HRMS) and liquid chromatography-high resolution mass spectrometry (LC-HRMS) with complementary instrumentation as well as nuclear magnetic resonance (NMR). These multiple platforms will allow us to capture an extended array of endogenous metabolites and to accurately define the related metabolic pathways. Data processing and analysis will be done using R packages and available databases.

Based on all of the above datasets (i.e., both the results of the toxicological assays described in the previous sections and the multi-omics data outlined above) an IATA as proposed by OECD will be developed in order to make the most efficient use of all available data and converge towards an ITS that couples advanced bioinformatics [39] and computational analysis of toxicological and biomonitoring data to targeted testing assays [40]. Our approach is to use computational systems biology to bridge, in an integrative way, findings from the advanced experimental (in vitro and in vivo) and in silico methodologies outlined above [41]. The systems biology track in OBERON focuses on interactions between elements (e.g., genes, proteins, and chemicals) at various biological levels (e.g., adipose tissue, liver, endocrine pancreas) integrating various data types such as biological endpoints (e.g., insulin and glucagon secretion for the endocrine pancreas) and ‘multi-omics’ data [42,43] towards the development of adverse outcome pathways (AOPs) that can be associated with exposure to putative EDs [44], thus enhancing the consequent human health risk assessment. Joint pathway analysis of multi-omics data will be performed using high-performance, advanced bioinformatics and visualization tools for multi-omics data interpretation. Systems biology-based AOP networks will be developed by (1) using artificial intelligence to explore existing knowledge [45,46], and (2) integrating the most relevant profiling and phenotypic data generated in relation to external exposure and health observations. AOP models will be linked to PBPK models to bridge external and internal exposure, and they will be parameterized using the omics data (development phase) from in vitro as well as human cohort data [47]. Developing systems biology networks and clusters offers the possibility to analyze, integrate and exploit optimally large-scale data, and combine them with available information such as high confidence human interactome, toxicogenomics data, biological pathways, AOPs and other existing data repositories [48,49,50,51]. To better understand pathways of toxicity (PoT), uncharacterized AOPs will be developed, and existing AOPs will be improved/optimized, i.e., MIEs or KE linked to exposure to putative EDs. AOPs are laid out in a linear fashion, progressing from the MIEs to the adverse outcome (AO) through various KEs via key event relationships (KERs). Recently, computational studies based on artificial intelligence have shown putative linkages between EDs (i.e., BPS and BPF), that are considered as stressors, and AOPs related to metabolism [45,46]. However, it cannot be assumed that the demonstration of a MIE will always lead to the AO. This will be addressed by investigating the dose-dependent activation of PoT rather than individual KEs. Joint pathway analysis of omics data coupled with phenotypic anchoring based on human cohort data will allow distinguishing adaptive responses from persistent and adverse effects. To use the AOP methodology for regulatory risk assessment, quantitative AOPs will need to be established and thresholds for MIE to trigger the downstream KEs or AOs need to be defined [39,52]. The construction of the IATA, which will follow the decision tree of OBERON will be continuously updated and optimized during the project, in order to provide an optimal, efficient and reliable tiered IATA strategy for EDs related to metabolic disorders for future consideration in guidelines (e.g., OECD). All in vitro, in vivo and in silico developed tests will be predictive tools. Each of them will provide ‘scores’, which will allow us to determine whether a substance is likely to be an ED targeting metabolism or not. The scores will depend on the technology employed, e.g., concentration of half-maximal activity (AC5O) or statistical p-value. The sensitivity (the capacity of a test to detect positive compounds) and the specificity (the capacity of a test to no detect negative compounds) of each in vitro test will be determined by testing a small chemical training set.

## 4. State of the Project and Expected Outcomes

The OBERON project has now started for a bit more than a year. Much of the effort was devoted to establishing the different experimental systems, taking into account ethical considerations. The systems are now running and some very preliminary experiments have been carried out. As an illustration, we performed a pilot study to assess the metabolomic profile of 20 human serum samples, and we are currently analyzing those initial results, and preparing the extension of the assays. Furthermore, we have prepared dedicated flyers explaining the relevance of metabolic outcomes for EDs.

Ultimately, the OBERON project will contribute to the improvement of current tiered approaches on metabolic outcomes of exposure to endocrine disruptors, expected to be operational in 2024, by:-Delivering an innovative screening battery based on new and robust tests combining experimental and computational strategies for ED-related metabolic disorders assessment, to support current OECD tiered approaches.-Providing a validated tiered IATA (i.e., a decision tree) ready for submission to JRC with the aim to reduce the use of animal testing in industries and regulatory agencies.-Improving mechanistic toxicological knowledge and integrating them into AOPs with the aim of improving risk assessment frameworks for human health effects. This will feed into the AOP-wiki and the OECD AOP approach.-Providing putative effect markers for metabolic disorders such as new evidence on the potential metabolic and metabolomic effects of EDs in human populations.

## Figures and Tables

**Figure 1 ijms-21-02988-f001:**
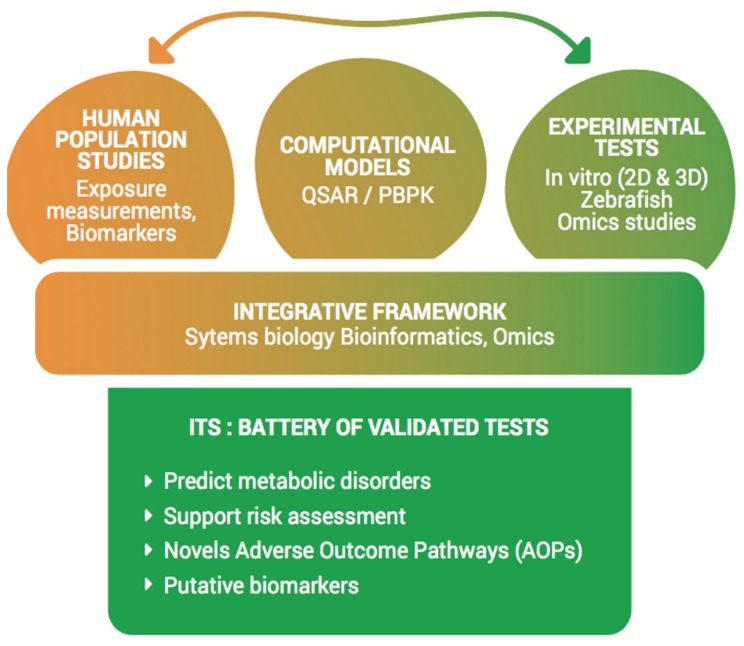
The OBERON platform. The OBERON project aims to establish an innovative integrated approach for testing and assessment (IATA) by providing a battery of effective and validated test systems for endocrine disruptor (ED) assessment on metabolic disorders. Multidisciplinary data and model systems, already existing or that will be developed within the OBERON consortium, will be integrated in order to perform predictive toxicology for EDs. Abbreviations: Quantitative Structure Activity Relationships (QSAR), Physiologically Based PharmacoKinetic (PBPK), Integrative Testing Strategy (ITS), Adverse Outcome Pathways (AOPs).

**Table 1 ijms-21-02988-t001:** The ten chemicals studied in OBERON.

Chemical Name	Acronym	CAS Number
Bisphenol A	BPA	80-05-7
Bisphenol S	BPS	80-09-1
Bisphenol F	BPF	620-92-8
Di(2-ethylhexyl) phthalate	DEHP	117-81-7
Dibutyl phthalate	DBP	84-74-2
Perfluorooctanesulfonic acid	PFOS	1763-23-1
Perfluorooctanoic acid	PFOA	335-67-1
Cadmium	Cd	7440-43-9
Dichlorodiphenyldichloroethylene	DDE	72-55-9
Butyl-paraben		94-26-8

**Table 2 ijms-21-02988-t002:** List of the human cohorts used in OBERON.

Cohort Name	Full Name and Key References	Country	Enrollment Period	No of Children at Birth
INMA	Environment and Childhood [9]	Spain	2004–2007	2000
HELIX	The human early-life exposome [10]	UK, Norway, Lithuania, France, Greece, Spain	1999–2010	1300
Pélagie	Endocrine disruptors: longitudinal study on pregnancy abnormalities, infertility, and childhood [11]	France	2002–2006	500
SEPAGES	Assessment of air pollution exposure during pregnancy and effect on health [12]	France	2014–2017	500
EXHES	European exposure and health examination survey (ISBN 978-3-319-89321-1)	Greece	2017–2019	300
ELSPAC	The European longitudinal study of pregnancy and childhood [13]	Czech Republic	1991–1992	300

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
