# Peer review of "Integrative Strategy of Testing Systems for Identification of Endocrine Disruptors Inducing Metabolic Disorders—An Introduction to the OBERON Project"

_ijms, 2020, doi:10.3390/ijms21082988_

Round 1
Reviewer 1 Report
In this paper the authors described the OBERON project, which aimed at assessing the toxicity of selected endocrine disrupting chemicals on metabolic endpoints through an integrated approach combining results from in vitro and in vivo experiments, omics technologies, epidemiological and biomonitoring studies, and modelling and computational systems biology. The paper is well written and interesting, given the widespread use of these compounds, the multiple routes of exposure and the uncertainties on their effects to humans, and provides new insights into test systems for identification of endocrine disruptors involved in metabolic disease, reducing use of animal tests and enhancing the development of alternative techniques not sufficiently exploited so far.
The comments below are in order of appearance.
A list of abbreviations should be added. Furthermore, please always provide the full name of abbreviations on first use throughout the text.
Lines 71-72. This statement requires a proper reference.
Lines 74-76. Please provide some examples.
Line 130. Consider changing “that” to “which”.
Figure 1. Please add a list of abbreviations used in a footnote.
Line 222. Consider using the paragraph as a bullet point.
Line 228. Please provide adequate references.
Line 257. Throughout the whole manuscript, the authors cite omics several times (see also lines 321-325; lines 390-391, 396), but without providing sufficient detail. Consider adding a brief explanation of the planned methods and techniques.
Lines 289-293. Please rephrase for better clarity.
Line 311. Please amend the typo in PPARγ.
Author Response
Thank you for your decision concerning our manuscript entitled “Integrative strategy of testing systems for identification of endocrine disruptors inducing metabolic disorders – An introduction to the OBERON project”. We have addressed all your comments and suggestions. We would like to thank you for the relevant comments, that allowed to improve our manuscript.
Please, find below in red, our responses.
The comments below are in order of appearance.
A list of abbreviations should be added. Furthermore, please always provide the full name of abbreviations on first use throughout the text.
We have added the full list of the used abbreviations, and checked that the full name of each abbreviation is mentioned at the first used.
Lines 71-72. This statement requires a proper reference. Done
Lines 74-76. Please provide some examples. Done
Line 130. Consider changing “that” to “which”. Done
Figure 1. Please add a list of abbreviations used in a footnote. The abbreviations have been added
Line 222. Consider using the paragraph as a bullet point. Done
Line 228. Please provide adequate references. Done
Line 257. Throughout the whole manuscript, the authors cite omics several times (see also lines 321-325; lines 390-391, 396), but without providing sufficient detail. Consider adding a brief explanation of the planned methods and techniques.
A description of the planned omics studies has been added on page 19.
Lines 289-293. Please rephrase for better clarity. Done
Line 311. Please amend the typo in PPARγ. Done
Reviewer 2 Report
In this paper, Audouze and coworkers describe the main concepts and ideas behind the OBERON project and the EURION cluster to which the project belongs. Specifically, OBERON is a 6.7M€-project coordinated by the French National Institute of Health and Medical Research (INSERM), featuring 11 European partners from 6 member states and funded by the H2020 Research and Innovation Program. OBERON has a multidisciplinary nature and it aims at developing experimental and computational standard procedures for the assessment of the metabolic outcomes following exposure to Endocrine Disruptor (ED) chemicals. Data generated in OBERON are expected to support current OECD and JRC testing procedures for EDs, to have predictive power towards new EDs and, ultimately, to be useful for regulatory purposes at European and global level.
I think this paper is of great interest since OBERON—and more generally EURION—is a very important European scientific initiative that is likely to greatly improve our current knowledge on EDs, possibly influencing future EU regulatory decisions.
I have just one “major concern” about it:
1 – Even if the declared focus of the paper is put on the description of the “OBERON vision” and related objectives, as a project report, this paper should provide (at least mention) the current state of project implementation. After 15 months from the kickoff (the project started on 1 January 2019), is any milestone of the project already (or about to be) achieved? How's work going?
Please, consider adding a section about it.
In addition, here are some minor issues:
1 – Although the Authors provide the link to OBERON website containing all the relevant information on the project, the essential temporal coordinates should be mentioned in the paper. Authors just says that the project started on 1 January 2019, but the end date (31 December 2023) should also be added to suggest the idea that OBERON pipeline is supposed to be fully operational by 2024.
2 – The report is clear and well written; however, the huge number of acronyms reduces somehow its readability. It would help, especially in order to expand the target audience, to provide a comprehensive list of the abbreviations used for both scientific (DEHP, BMI, WAT,…) and institutional (EC, OECD, JRC, DG…) entities.
3 – A couple of typos:
Lines 101-102:
report on "setting priorities: https://op.europa.eu...
Should be
report on “setting priorities”: https://op.europa.eu...
or
report on DG Environment workshop: https://op.europa.eu...
Line 311:
PPARg
Should be
PPARγ
Author Response
Thank you for your decision concerning our manuscript entitled “Integrative strategy of testing systems for identification of endocrine disruptors inducing metabolic disorders – An introduction to the OBERON project”. We have addressed all comments and suggestions made by the reviewers. We would like to thank them for their relevant comments, that allowed to improve our manuscript.
We would hereby like to re-submit the manuscript to be considered for publication as a report project in the special issue “Advances in the Research of Endocrine Disrupting Chemicals 2.0" Please, find below in red, our responses.
I have just one “major concern” about it:
1 – Even if the declared focus of the paper is put on the description of the “OBERON vision” and related objectives, as a project report, this paper should provide (at least mention) the current state of project implementation. After 15 months from the kickoff (the project started on 1 January 2019), is any milestone of the project already (or about to be) achieved? How's work going?
Please, consider adding a section about it.
We have added a part under the section 4, that mention the status of the project. As the present manuscript is a project report, we haven’t added more results.
In addition, here are some minor issues:
1 – Although the Authors provide the link to OBERON website containing all the relevant information on the project, the essential temporal coordinates should be mentioned in the paper. Authors just says that the project started on 1 January 2019, but the end date (31 December 2023) should also be added to suggest the idea that OBERON pipeline is supposed to be fully operational by 2024.
We have modified the text accordingly.
2 – The report is clear and well written; however, the huge number of acronyms reduces somehow its readability. It would help, especially in order to expand the target audience, to provide a comprehensive list of the abbreviations used for both scientific (DEHP, BMI, WAT,…) and institutional (EC, OECD, JRC, DG…) entities.
We have added the full list of the used abbreviations, and checked that the full name of each abbreviation is mentioned at the first used.
3 – A couple of typos:
Lines 101-102:
report on "setting priorities: https://op.europa.eu...
Should be
report on “setting priorities”: https://op.europa.eu...
or
report on DG Environment workshop: https://op.europa.eu...
We have corrected it.
Line 311:
PPARg
Should be
PPARγ Done
Reviewer 3 Report
The study appears to have been well-planned and executed. There are only minor comments to be corrected. This reviewer suggests that the manuscript is acceptable in the current version.
Line 55: endocrine-disrupting
Line 62: 150 billion
Line 65: double this amount
Line 66: to protect
Line 68: and member
Line 69: to ensure
Line 73: a risk
Line 74: some cases
Line 81: species-dependent metabolic pathways and the
Line 82: New developments
Line 86: Evaluation, and
Line 88: A Vision
Line 94: the identification
Line 100: the JRC
Line 116: their objectives,
Line 126: , and
Line 143: Table 1
Line 148: , and systems biology, will allow improving
Line 149: the development
Line 165: OBERON aims
Line 175: a larger
Line 189: in the Czech
Line 191: in EDs exposure
Line 192: Currently, tests for predicting human toxicity of EDs are based on animal studies, which do
Line 197: studies and
Line 214: a large
Line 218: to be easily
Line 220: modeling
Line 224: via a nuclear
Line 226: endocrine-disrupting
Line 229: and lipid
Line 234: humans
Line 240: of prime
Line 271: tests and
Line 272: , as well as
Line 283: , and white
Line 286: following the toxicokinetic prediction of tissue-level concentrations
Line 311: and the absence
Line 312: allow us
Line 321: provide a greater
Line 351: The integration
Line 355: distribution, and
Line 364: to the developmental
Line 366: during the early
Line 368: prenatal life
Line 373: reconstructing
Line 374: points, especially at critical periods
Line 385: , proteins, and
Line 387: secretion for the endocrine
Line 405: the demonstration of a MIE
Line 407: KEs. Joint pathway analysis
Line 408: allow distinguishing
Line 413: related to metabolic
Line 416: half-maximal activity (AC5O)
Author Response
Thank you for your decision concerning our manuscript entitled “Integrative strategy of testing systems for identification of endocrine disruptors inducing metabolic disorders – An introduction to the OBERON project”. We have addressed all your comments and suggestions. We would like to thank you for the relevant comments, that allowed to improve our manuscript.
Please, find below in red, our responses.
The study appears to have been well-planned and executed. There are only minor comments to be corrected. This reviewer suggests that the manuscript is acceptable in the current version.
Line 55: endocrine-disrupting
Line 62: 150 billion
....
We have modified all the suggested points.
Reviewer 4 Report
The manuscript is suitable for puplishing. The report introduced the OBERON project methodology and outcomes that will deliver screening battery based on integration of in vivo & in vitro experimental and computational stratigies for endocrine disruptors including metabolic disorders assessment , with the aim to improve risk assessment frameworks for human health effects .
Author Response
We thank the reviewer for her/his positive comment.
Round 2
Reviewer 1 Report
The authors addressed my previous comments and not the manuscript is suitable for publication.
Reviewer 2 Report
The authors have satisfactorily addressed all my concerns and made the necessary changes to the manuscript.
Some very minor typos remain to be corrected:
line 472-473 (Section 4):
Much of the effort was devoted to establish the different…
Should be
Much of the effort was devoted to establishing the different…
In “Abbreviations” list:
PPARA γ: peroxisome proliferator-activated receptor gamma
Should be
PPARγ: peroxisome proliferator-activated receptor gamma